# Balanced Quantum-Like Bayesian Networks

**DOI:** 10.3390/e22020170

**Published:** 2020-02-02

**Authors:** Andreas Wichert, Catarina Moreira, Peter Bruza

**Affiliations:** 1Department of Computer Science and Engineering, INESC-ID & Instituto Superior Técnico, University of Lisbon, 2740-122 Porto Salvo, Portugal; andreas.wichert@tecnico.ulisboa.pt; 2School of Information Systems, Science and Engineering Faculty, Queensland University of Technology, QLD 4000 Brisbane, Australia; p.bruza@qut.edu.au

**Keywords:** decision making, quantum cognition, quantum-like Bayesian networks, law of total probability, probability waves

## Abstract

Empirical findings from cognitive psychology indicate that, in scenarios under high levels of uncertainty, many people tend to make irrational decisions. To address this problem, models based on quantum probability theory, such as the quantum-like Bayesian networks, have been proposed. However, this model makes use of a Bayes normalisation factor during probabilistic inference to convert the likelihoods that result from quantum interference effects into probability values. The interpretation of this operation is not clear and leads to extremely skewed intensity waves that make the task of prediction of these irrational decisions challenging. This article proposes the law of balance, a novel mathematical formalism for probabilistic inferences in quantum-like Bayesian networks, based on the notion of balanced intensity waves. The general idea is to balance the intensity waves resulting from quantum interference in such a way that, during Bayes normalisation, they cancel each other. With this representation, we also propose the law of maximum uncertainty, which is a method to predict these paradoxes by selecting the amplitudes of the wave with the highest entropy. Empirical results show that the law of balance together with the law of maximum uncertainty were able to accurately predict different experiments from cognitive psychology showing paradoxical or irrational decisions, namely in the Prisoner’s Dilemma game and the Two-Stage Gambling Game.

## 1. Introduction

This article proposes a novel type of probabilistic inference in quantum-like Bayesian networks [1] based on the notion of intensity waves. We refer to intensity waves the principle of an electron being represented as a wave under uncertainty: if one does not perform any measurement, then the electron enters into a superposition state and takes the properties of a wave, which evolves through time and which can generate quantum interference effects. The underlying core idea of the proposed model is to provide a novel mathematical formalism that, under probabilistic inference in quantum-like Bayesian networks, will enable the intensity waves to cancel each other during Bayes normalisation. This will result in probabilistic waves that are balanced, contrary to the current approach, where they are skewed in order to meet Bayes normalisation. These intensity waves are highly significant in the literature, because they can provide means to quantify uncertainty during probabilistic inferences, and consequently the prediction of inferences that are either obeying or violating the rules of classical probability theory.

The main motivation of proposing such a model is based on the fact that empirical evidence from the cognitive psychology literature suggest that most human decision-making cannot be adequately modelled using classical probability theory as defined by Kolmogorov’s axioms [2,3,4,5,6]. These empirical findings show that, under uncertainty, humans tend to violate the expected utility theory and consequently the laws of classical probability theory (e.g., the law of total probability [7]), leading to what is known as the “disjunction effect” which, in turn, leads to violation of the Sure Thing Principle.

The concept of human behaviour deviating from the predictions of expected utility theory is not something new. There are many studies suggesting that humans tend to deviate from optimal Bayesian decisions [8,9,10,11], which is not consistent with expected utility theory. Many approaches have been proposed in the literature in order to overcome these limitations. For instance, recently Peters [12] criticised the notion of expected utility by showing that it is built under false assumptions, and proposed the concept of ergodicty economics as an alternative model to optimise time-average growth rates. Another work from Schwartenbeck et al. [13] describes irrational behaviour as a characteristic of suboptimal behaviour of specific groups [14]. Their main argument is to look at irrational behaviour as a subject-specific generative model of a task, as opposed of having a single optimal model of behaviour (like it is predicted by expected utility theory). Other models have been recently proposed in the literature based on a more general probabilistic framework: the formalism of quantum probability theory, which had led to the emergence of a new research field called quantum cognition [7,15,16,17,18,19,20,21].

The main difference between classical and quantum probability theory lies in the fact that in classical probability all properties of events are assumed to be in a definite state. As a consequence, all properties are assumed to have a definite value before measurement, and that this value is the outcome of the measurement [22]. In quantum probability theory, on the other hand, all properties of events are in an indefinite state prior to measurement and are represented by a wave function, which evolves in a smooth and continuous way according to Schrödinger’s equation. This evolution is deterministic and reversible and is done in parallel. Reversible means that no information is lost. This kind of evolution is described by quantum probabilities that are also called von Neumann probabilities. However, during the observation (measurement), the wave collapses into a definite state. The subsequent observed evolution is described by Kolmogorov’s probabilities, which are neither smooth, nor reversible, since information is lost. In terms of probabilistic inference in quantum-like models, this suggests that probabilities are composed of two terms: one corresponding to the outcome of Kolmogorov’s (classical) probabilities, and another which corresponds to the interference effects between the intensity waves that occur before measurement.

Current models for probabilistic inference in quantum-like Bayesian networks apply Bayes normalisation factor to both the classical terms and interference terms. While it is clear the application of Bayes normalisation to classical outcomes, it is not clear what is the interpretation when it comes to its application to quantum interference terms. One can argue that this is just a way to normalise likelihoods in order to convert them into probabilities, however the resulting intensity waves lack interpretation and become extremely skewed, resulting in a representation that is very sensitive to initial conditions of amplitudes. This can lead to some significant challenges, since one core research question is how to quantify the decision-maker’s uncertainty during probabilistic inferences using the amplitudes of the intensity waves, in such a way that it can predict probabilistic outcomes which are either following a definite Kolmogorovian setting, or disjunction effects, or other violations to the laws of classical probability.

To address these challenges this article extends the initial work conducted in Wichert and Moreira [23], and investigates the relationship between classical and quantum probabilities and we propose two laws that will allow the representation of the intensity waves. The law of balance which is a way to normalise intensity waves without the need of using Bayes normalisation factor, and the law of maximum uncertainty that enables the quantification of uncertainty within the paradigm of these intensity waves. In short, this article contributes:A Law of Balance: a novel mathematical formalism for quantum-like probabilistic inferences that enables the cancellation of quantum interference terms upon the application of Bayes normalisation factor. This way, the amplitudes of the probability waves become balanced.A Law of Maximum Uncertainty: which states that in order to predict disjunction effects, one should choose the amplitude of the wave that contains the maximum uncertainty or the the maximum information.

These laws are validated in quantum-like probabilistic inferences in Bayesian networks in cognitive psychology experiments from the literature, namely the Prisoner’s Dilemma game [24,25,26,27] and the Two Stage Gambling Game [28,29].

We would like to highlight that the purpose of the present paper is not to say that quantum cognition is the best approach to fully understand human behaviour. We see quantum cognition as an approach which is as promising as other approaches in the literature for this end with its advantages and disadvantages. The main goal of this work is simply to continue to develop quantum-like models for cognition, since they still suffer from many gaps, one of them is precisely on how to deal with quantum interference during probabilistic reasoning and how that could be applied in more general structures, not only for cognition, but for general decision-making models.

## 2. Probabilistic Inference in Bayesian Networks

In this section, we present the fundamental concepts regarding probabilistic inference within Bayesian networks. Bayesian Networks are directed acyclic graphs in which each node represents a random variable from a specific domain, and each edge represents a direct influence from the source node to the target node. The graph represents independence relationships between variables, and each node is associated with a conditional probability table that specifies a distribution over the values of a node given each possible joint assignment of values of its parents [30].

In these networks, the graphical relationship between random variables is fundamental to determine conditional independence and to compute probabilistic inferences. For instance, consider two events represented by the binary random variables, *X* and *Y*, which can either be true ({X=x,Y=y}) or false ({X=¬x,Y=¬y}. For simplicity, throughout this paper, we will refer to p(X=x) as p(x), p(Y=y) as p(y), p(X=¬x) as p(¬x) and p(Y=¬y) as p(¬y). This translates into
(1)p(x)+p(¬x)=1,p(y)+p(¬y)=1,
where the law of total probability is given by
(2)p(y)=p(y,x)+p(y,¬x).
Using the chain rule,
p(y)=p(y|x)p(x)+p(y|¬x)p(¬x).
The same relationship is obtained for p(¬y),
(3)p(¬y)=p(¬y,x)+p(¬y,¬x).

This probabilistic influence that random variables *X* exerts *Y* can be represented by the Bayesian network depicted in Figure 1.

If two variables *x* and *y* are independent, then the probability that the event *x* and *y* simultaneously occur is
(4)p(x,y)=p(x∧y)=p(x)p(y).

For *N* independent random variables, we obtain
(5)$$p\left(x_{1},x_{2},\cdots ,x_{N}\right)=\prod_{i=1}^{N}p\left(x_{i}\right).$$
When not all events are independent, then we can decompose the probabilistic domain into subsets via conditional independence by using Bayes’ rule,
(6)$$p\left(x_{1}|x_{2}\right)=\frac{p(x_{1},x_{2})}{p(x_{2})}=\frac{p\left(x_{2}|x_{1}\right)p\left(x_{1}\right)}{p(x_{2})}.$$
This translates in the chain rule formula. Assuming that $x_{2}$ and $x_{3}$ are independent, but $x_{1}$ is conditionally dependent given $x_{2}$ and $x_{3}$, then
(7)$$p\left(x_{1},x_{2},x_{3}\right)=p\left(x_{1}|x_{2},x_{3}\right)p\left(x_{2}\right)p\left(x_{3}\right).$$
Analogously, if we assume that $x_{4}$ is conditionally dependent given $x_{1}$ but independent of $x_{2}$ and $x_{3}$, then
$$p\left(x_{1},x_{2},x_{3},x_{4}\right)=p\left(x_{1}|x_{2},x_{3}\right)p\left(x_{2}\right)p\left(x_{3}\right)p\left(x_{4}|x_{1}\right)p\left(x_{4}|x_{1}\right).$$

From this representation, it follows a causal relationship between these events, which is represented by a conditional dependence. Figure 2 shows a graphical representation indicating the causal influence between events $x_{1}$, $x_{2}$, $x_{3}$ and $x_{4}$.

In our example $x_{2}$ and $x_{3}$ cause $x_{1}$ and $x_{1}$ also causes $x_{4}$. This kind of decomposition via conditional independence is modelled by Bayesian networks, since they provide a natural representation for (causally induced) conditional independence. These conditional independence assumptions are also represented by the topology of an acyclic directed graph and by sets of conditional probabilities. In the network each variable is represented by a node and the links between them represent the conditional independence of the variable towards its non descendants and its immediate predecessors (see Figure 2).

The corresponding network topology reflects our belief in the associated causal knowledge. Consider the well-known example of Judea Perl [31,32]. “I am at work in Los Angeles, and neighbour John calls to say that the alarm of my house is ringing. Sometimes minor earthquakes set off the alarm. Is there a burglary?” Constructing a Bayesian network should be easy, because each variable is directly influenced by only a few other variables. In the example, there are four variables, namely, $Burglary(=x_{2}$), $Earthquake(=x_{3}$), $Alarm(=x_{1}$) and $JohnCalls(=x_{4}$). Due to simplicity, we ignore an additional variable MaryCalls that was present in the original example. The corresponding network topology in Figure 2 reflects the following “causal” knowledge:A burglar can set the alarm on.An earthquake can set the alarm on.The alarm can cause John to call.

Bayesian networks represent for each variable a conditional probability table which describes the probability distribution of a specific variable given the values of its immediate predecessors. A conditional distribution for each node $x_{i}$ given its parents is
$$p(x_{i}|Parent_{1}\left(x_{i}\right),Parent_{2}\left(x_{i}\right),\ldots ,Parent_{k}\left(x_{i}\right)),$$
with *k* representing the number of predecessor nodes (or parent nodes) of node $x_{i}$. Given the query variable *x* whose value has to be determined and the evidence variable *e* which is known and the remaining unobservable variables *y*, we perform a summation over all possible *y*. In the following examples, for simplification the variables are binary and describe binary events. All possible values (true/false) of the unobservable variables *y* are determined according to the law of total probability
(8)$$p\left(x|e\right)=\alpha \sum_{y}p\left(x,e,y\right)=\alpha \left(p\left(x,e,y\right)+p\left(x,e,\neg y\right)\right).$$
with
$$\alpha =\frac{1}{p(e)}=\frac{1}{\sum_{y}p\left(x,e,y\right)+\sum_{y}p\left(\neg x,e,y\right)}.$$

## 3. Quantum Probabilities

Until “recently” quantum physics was the only branch in science that evaluated a probability p(x) of a state *x* as the squared magnitude of a probability amplitude A(x), which is represented by a complex number
(9)$$p\left(x\right)={\left|A\left(x\right)\right|}^{2}=A{\left(x\right)}^{*}A\left(x\right).$$
This is because the product of a complex number with its conjugate is always a real number. With
(10)A(x)=α+βi
(11)$$A{\left(x\right)}^{*}A\left(x\right)=\left(\alpha -\beta i\right)\left(\alpha +\beta i\right)$$
(12)$$A{\left(x\right)}^{*}A\left(x\right)=\alpha^{2}+\beta^{2}={\left|A\left(x\right)\right|}^{2}.$$

Quantum physics by itself does not offer any justification or explanation beside the statement that it just works fine, see Binney and Skinner [33].

## 4. The Two-Slit Experiment, Intensity Waves and Probabilisitc Waves

Suppose there are two mutual exclusive events *x* and *y*. This means that *x* and *y* do not occur together.

The classical probability of an event *x* or event *y* is just
(13)p(x∨y)=p(x)+p(y).
This is the sum rule for probabilities for exclusive events. For probability amplitudes, it is as well
(14)A(x∨y)=A(x)+A(y).

However converting these amplitudes into probabilities according to Equation (Equation 9) leads to an interference term $2ℜ(A\left(x\right)A^{*}\left(y\right))$,
(15)$$\begin{array}{c} {\left|A\left(x\right)+A\left(y\right)\right|}^{2}={\left|A\left(x\right)\right|}^{2}+{\left|A\left(y\right)\right|}^{2}+2ℜ\left(A\left(x\right)A^{*}\left(y\right)\right)\\ \;=p\left(x\right)+p\left(y\right)+2ℜ(A\left(x\right)A^{*}\left(y\right)), \end{array}$$
making both approaches, in general, incompatible
(16)$${\left|A\left(x\right)+A\left(y\right)\right|}^{2}\neq p\left(x\right)+p\left(y\right).$$

In other words, the summation rule of classical probability theory is violated, resulting in one of the most fundamental laws of quantum mechanics, see [33]. In the following sections, rather than dealing with binary events, we will introduce the notion of a “state” which corresponds to some states of nature. Logical possibilities of events are usually called the elementary events or states of nature. Here we will refer to them simply as “states”.

The relation between the amplitudes and probabilities in quantum theory is related to an unobservable wave function. The wave function in quantum mechanics represents a superposition of states of which each state *x* is represented by A(x). Suppose that an unobservable state evolves smoothly and continuously. However, during the measurement, it collapses into a definite state with a probability $p\left(x\right)={\left|A\left(x\right)\right|}^{2}$. For instance, let us imagine a gun that fires electrons and a screen with two narrow slits *x* and *y* and a photographic plate. An emitted electron can pass through slit *x* or slit *y* and reaches the photographic plate at the position *z*, which is equidistant from both slits. The electron detectors show from which slit the electron went through, and we find that the probability of the electron hitting the photographic plate is
(17)p(z)=p(x)+p(y).
This probability means that, when measured, the electron behaved as a particle.

### 4.1. Intensity Waves

On the other hand, if we remove the detectors, the electron is unobserved, not knowing through which slit it went through. Now, the electron is represented as a wave with the amplitudes
(18)$$a\left(x,\theta_{1}\right)=\sqrt{p(x)}e^{i\theta_{1}}=A\left(x\right),$$
(19)$$a\left(y,\theta_{2}\right)=\sqrt{p(y)}e^{i\theta_{2}}=A\left(y\right).$$
These amplitudes contain a parameter θ, which corresponds to the phase of the wave. The equation then becomes
(20)$$I\left(z,\theta_{1},\theta_{2}\right)={\left|a\left(x,\theta_{1}\right)+a\left(y,\theta_{2}\right)\right|}^{2}=\left(a\left(x,\theta_{1}\right)+a\left(y,\theta_{2}\right)\right){\left(a\left(x,\theta_{1}\right)+a\left(y,\theta_{2}\right)\right)}^{*}$$
with
$$I\left(z,\theta_{1},\theta_{2}\right)\neq p\left(z\right)$$
for most values of $\theta_{1},\theta_{2}$. Since the value of $I(z,\theta_{1},\theta_{2})$ may be bigger than one, we can not identify it with probability values. We call $I(z,\theta_{1},\theta_{2})$ the intensity wave of the state *z*. Since the norm is being positive or more precisely non-negative, the intensity wave of the state *z*,

$I(z,\theta_{1},\theta_{2})$, is always non-negative
(21)$$0\leq I\left(z,\theta_{1},\theta_{2}\right)={∥a\left(x,\theta_{1}\right)+a\left(y,\theta_{2}\right)∥}^{2}.$$
It follows
$$I\left(z,\theta_{1},\theta_{2}\right)=\left(\sqrt{p(x)}e^{i\theta_{1}}+\sqrt{p(y)}e^{i\theta_{2}}\right)\left(\sqrt{p(x)}e^{-i\theta_{1}}+\sqrt{p(y)}e^{-i\theta_{2}}\right)$$
$$I\left(z,\theta_{1},\theta_{2}\right)=p\left(x\right)+p\left(y\right)+\sqrt{p(y)p(x)}e^{i(\theta_{2}-\theta_{1})}+\sqrt{p(x)p(y)}e^{i(\theta_{1}-\theta_{2})}$$
(22)$$I\left(z,\theta_{1},\theta_{2}\right)=p\left(x\right)+p\left(y\right)+\sqrt{p(x)p(y)}\left(e^{i(\theta_{1}-\theta_{2})}+e^{-i(\theta_{1}-\theta_{2})}\right).$$
With
(23)$$\cos\left(\theta_{1}-\theta_{2}\right)=\frac{\left(e^{i(\theta_{1}-\theta_{2})}+e^{-i(\theta_{1}-\theta_{2})}\right)}{2},$$
we get
(24)$$I\left(z,\theta_{1},\theta_{2}\right)=p\left(x\right)+p\left(y\right)+2\sqrt{p(x)p(y)}\cos\left(\theta_{1}-\theta_{2}\right).$$
At different positions at the photographic plate an interference pattern emerges due to the different phase values $\theta_{1}-\theta_{2}$ that change with time. They are non-constant contrary to the values *x* and *y*. This corresponds the wave-particle duality that states that all matter exhibits both wave and particle properties. For binary events,
(25)p(x)+p(¬x)=1,p(y)+p(¬y)=1,
the law of total quantum probability corresponds to the intensity waves
(26)$$I\left(y,\theta_{1},\theta_{2}\right)={\left|a\left(y,x,\theta_{1}\right)+a\left(y,\neg x,\theta_{2}\right)\right|}^{2}.$$
For simplification we can replace $\theta_{1}-\theta_{2}$ with θ,
$$\theta =\theta_{1}-\theta_{2},$$
(27)$$I\left(y,\theta \right)=p\left(y\right)+2\sqrt{p(y,x)p(y,\neg x)}\cos\left(\theta \right)$$
and
(28)$$I\left(\neg y,\theta_{\neg 1},\theta_{\neg 2}\right)={\left|a\left(\neg y,x,\theta_{\neg 1}\right)+a\left(\neg y,\neg x,\theta_{\neg 2}\right)\right|}^{2}$$
with
$$\theta_{\neg }=\theta_{\neg 1}-\theta_{\neg 2}$$
(29)$$I\left(\neg y,\theta_{\neg }\right)=p\left(\neg y\right)+2\sqrt{p(\neg y,x)p(\neg y,\neg x)}\cos\left(\theta_{\neg }\right)$$
and for certain phase values
(30)$$I\left(y,\theta \right)+I\left(\neg y,\theta_{\neg }\right)\neq 1.$$
In Figure 3a we see two intensity waves in relation to the phase with the parametrisation as indicated in the Bayesian network represented in Figure 1.

### 4.2. Probability Waves

Intensity waves I(y,θ) and $I(\neg y,\theta_{\neg })$ are probability waves p(y,θ) and $p(\neg y,\theta_{\neg })$ if:They sum to one
(31)$$p\left(y,\theta \right)+p\left(\neg y,\theta_{\neg }\right)=p\left(y\right)+p\left(\neg y\right)=1;$$They are bigger or equal than 0 and smaller or equal than one
(32)$$0\leq p\left(y,\theta \right)\leq 1,\;\;0\leq p\left(\neg y,\theta_{\neg }\right)\leq 1.$$

### 4.3. Normalisation

During probabilistic inference in quantum-like Bayesian networks, normalisation is done in the following way (see [1,34,35]):(33)$$p\left(y,\theta \right)=\frac{I(y,\theta )}{I\left(y,\theta \right)+I\left(\neg y,\theta_{\neg }\right)},$$
and
(34)$$p\left(\neg y,\theta_{\neg }\right)=\frac{I(\neg y,\theta_{\neg })}{I\left(y,\theta \right)+I\left(\neg y,\theta_{\neg }\right)}$$
with
(35)$$p\left(y,\theta \right)+p\left(\neg y,\theta_{\neg }\right)=1.$$

The probability waves collapse to the classical setting, when the interference term is 0. In other words, when $\theta =\theta_{\neg }=\pm \frac{\pi }{2}$. Usually it is assumed that $\theta =\theta_{\neg }$.
$$p\left(y\right)=p\left(y,\pm \frac{\pi }{2}\right),p\left(\neg y\right)=p\left(\neg y,\pm \frac{\pi }{2}\right)$$

Figure 3b shows the two probability waves p(yθ) and $p(\neg y,\theta_{\neg })$ in relation to the Bayesian network presented in Figure 1. Using this normalisation formula, one can see that the probability waves become very skewed and extremely sensitive to changes in the waves’ θ parameter. A more balanced representation of the intensity waves would be beneficial for quantum-like Bayesian networks, to overcome this *deterministic chaos* that was pointed out in Moreira and Wichert [1].

We will now place the preceding conceptual framework and associated formalism from quantum mechanics within the context of human decision-making. As stated previously, human decision-making may violate the law of total probability and indicate a subjective probability $p_{sub}\left(y\right)$. In quantum cognition, the probability wave p(y,θ) is used to model the subjective probability with a value $p_{q}\left(y\right)$ with
$$p_{sub}\left(y\right)=p_{q}\left(y\right),\;\;p_{q}\left(y\right)=p\left(y,\theta^{*}\right).$$

Usually the value of $\theta^{*}$ is manually fitted for each case, see [35]. A disadvantage of individual fitting is that it does not allow the possibility to make predictions or frame generalisations.

In Moreira and Wichert [1] a dynamic heuristic was proposed in which thresholds are determined by learning from the data from a certain domain. In this work, we investigate if there is a straightforward meaningful relationship between the phase, θ and the resulting probability $p_{q}\left(y\right)$ that could explain $p_{sub}\left(y\right)$. This leads to our first contribution, the *Law of Balance*.

## 5. The Law of Balance

Instead of the simple Bayes normalisation of the intensity waves, we propose a novel normalisation technique based on the law of balance.

In the law of balance, the interference between the two waves is balanced, which means that the interference term of p(y,θ) and the interference term $p(\neg y,\theta_{\neg })$ cancel each out during probabilistic inference. In other words,
(36)$$\sqrt{p(y,x)p(y,\neg x)}\cos\left(\theta \right)=-\sqrt{p(\neg y,x)p(\neg y,\neg x)}\cos\left(\theta_{\neg }\right).$$

We can solve the Equation (Equation 36) in terms of the phase $\theta_{\neg }$ or θ, which results in the three possible cases, which are the core of the proposed Law of Balance.
**Case 1: probability wave**$p(\neg y,\theta_{\neg })$**dominates probability wave**p(y,θ). For the constraint
(37)$$\frac{p(y|x)p(y|\neg x)}{p(\neg y|x)p(\neg y|\neg x)}<1,$$
then, we get that the probability wave $p(\neg y,\theta_{\neg })$ dominates (or is bigger than) the probability wave p(y,θ). This means that the probability wave $p(\neg y,\theta_{\neg })$ determines the other wave as
(38)$$p\left(\neg y,\theta_{\neg }\right)=1-p\left(y,\theta \right),$$
where
(39)$$\theta_{\neg }=cos^{-1}\left(-\sqrt{\frac{p(y|x)p(y|\neg x)}{p(\neg y|x)p(\neg y|\neg x)}}\cos\left(\theta \right)\right).$$**Case 2: probability wave**p(y,θ)**dominates probability wave**$p(\neg y,\theta_{\neg })$.For the constraint,
(40)$$\frac{p(\neg y|x)p(\neg y|\neg x)}{p(y|x)p(y|\neg x)}\leq 1.$$
then, we get that the probability wave p(y,θ) dominates (or is bigger than) the probability wave $p(\neg y,\theta_{\neg })$. This means that the probability wave p(y,θ) determines the other wave as
(41)$$p\left(y,\theta \right)=1-p\left(\neg y,\theta_{\neg }\right),$$
where
(42)$$\theta =\cos^{-1}\left(-\sqrt{\frac{p(\neg y|x)p(\neg y|\neg x)}{p(y|x)p(y|\neg x)}}\cos\left(\theta_{\neg }\right)\right)$$**Case 3: none of the waves dominate each other.**For the constraint
(43)$$\frac{p(y|x)p(y|\neg x)}{p(\neg y|x)p(\neg y|\neg x)}=\frac{p(\neg y|x)p(\neg y|\neg x)}{p(y|x)p(y|\neg x)}=1.$$
we get
$$\theta_{\neg }=\cos^{-1}\left(-\cos(\theta )\right)$$
(44)$$\theta_{\neg }=\cos^{-1}\left(\cos(\theta \pm \pi )\right)$$
or
$$\theta =\cos^{-1}\left(-\cos(\theta_{\neg })\right)$$
(45)$$\theta =\cos^{-1}\left(\cos(\theta_{\neg }\pm \pi \right)$$
This case applies for double stochastic models, like the ones proposed in [7,15,16,17,18].

From the constraints derived from the law of balance, one can easily prove that this law is in accordance to the axioms of probability theory, where the probability of an event is a non-negative real number smaller or equal to one, and that it is also in accordance with the current double stochastic models proposed in the literature, namely Busemeyer et al. [7], Busemeyer and Trueblood [17].
**Probability Waves are Non-Negative Real Numbers Smaller Equal One.**We assume without loss of generality that p(y)≤p(¬y). It follows that p(y)≤0.5. By the inequality of arithmetic and geometric means, we get the relationship
(46)$$\sqrt{p(y,x)p(y,\neg x)}\leq \frac{p(y,x)+p(y,\neg x)}{2}$$
(47)$$2\sqrt{p(y,x)p(y,\neg x)}\leq p\left(y,x\right)+p\left(y,\neg x\right)$$
and
$$p\left(y,\theta \right)=p\left(y\right)+2\sqrt{p(y|x)p(x)p(y|\neg x)p(\neg x)}\cos\left(\theta \right)$$
resulting in the confirmation that probability waves are always smaller or equal than 1,
(48)p(y,θ)≤2p(y)≤1.**Conformity with Double Stochastic Models.**Double stochastic models correspond to the situation where condition 3 occurs, that is,
(49)$$\theta_{\neg }=\cos^{-1}\left(\cos(\theta \pm \pi )\right)\;\mathrm{or}\;\;\;\theta =\cos^{-1}(\cos(\theta_{\neg }\pm \pi ))$$
This means that, for instance, for θ=0, we have $\theta_{\neg }=\pi$ with
$$e^{i\theta }=e^{i0}=1\;\;\;\mathrm{and}\;\;\;e^{i\theta_{\neg }}=e^{i\pi }=-1.$$
Consequently, we obtain the unitary matrix
(50)$$\left(\begin{array}{cc} \sqrt{p(y|x)} & \sqrt{p(y|\neg x)}\\ \sqrt{p(\neg y|x)} & -\sqrt{p(\neg y|\neg x)} \end{array}\right)\left(\begin{array}{cc} \sqrt{p(y|x)} & \sqrt{p(y|\neg x)}\\ \sqrt{p(\neg y|x)} & -\sqrt{p(\neg y|\neg x)} \end{array}\right)=\left(\begin{array}{cc} 1 & 0\\ 0 & 1 \end{array}\right)$$
and for θ¬=±π it is θ=0 with the unitary matrix
(51)$$\left(\begin{array}{cc} \sqrt{p(y|x)} & -\sqrt{p(y|\neg x)}\\ \sqrt{p(\neg y|x)} & \sqrt{p(\neg y|\neg x)} \end{array}\right)\left(\begin{array}{cc} \sqrt{p(y|x)} & \sqrt{p(y|\neg x)}\\ -\sqrt{p(\neg y|x)} & \sqrt{p(\neg y|\neg x)} \end{array}\right)=\left(\begin{array}{cc} 1 & 0\\ 0 & 1 \end{array}\right)$$
Since the outcomes of Equations (Equation 50) and (Equation 51) are equal, then one can see the law of balance is is accordance with the unitary operators for double stochastic models [7,18].

## 6. Balanced Probability Waves

In this section, we investigate the boundaries that the phase parameter, θ and $\theta_{\neg }$ can have, in order to obtain probability waves that do not lose information.

According to the law of balance, the probability waves $p(\neg y,\theta_{\neg })$ and p(y,θ) are defined as
(52)$$p\left(\neg y,\theta_{\neg }\right)=p\left(\neg y\right)+2\sqrt{p(\neg y|x)p(x)p(\neg y|\neg x)p(\neg x)}\cos\left(\theta_{\neg }\right),$$
(53)$$p\left(y,\theta \right)=p\left(y\right)+2\sqrt{p(y|x)p(x)p(y|\neg x)p(\neg x)}\cos\left(\theta \right).$$

In Figure 3c, one can see two probability waves in relation to the phase with the parametrisation as indicated in the Bayesian network represented in Figure 1.
For p(y)≤p(¬y), the maximum interference is
(54)$$\pm Interference_{max}=\pm \sqrt{p(y,x)p(y,\neg x)}.$$For p(¬y)≤p(y), the maximum interference is
(55)$$\pm Interference_{max}=\pm \sqrt{p(\neg y,x)p(\neg y,\neg x)}.$$

We can define the intervals that describe the probability waves by
(56)$$I_{y}=\left[p\left(y\right)-Interference_{max},p\left(y\right)+Interference_{max}\right],$$
(57)$$I_{\neg y}=\left[p\left(\neg y\right)-Interference_{max},p\left(\neg y\right)+Interference_{max}\right]$$
with
(58)$$p\left(\neg y,\theta_{\neg }\right)\in I_{\neg y},\;\;p\left(y,\theta \right)\in I_{y}.$$

From these boundaries, an immediate question arises: How can we choose the phase θ without losing information about the probability waves?

According to the Bayesian network in Figure 1, with only two random variables, a wave is fully described by the probability value p(y) or p(¬y) and the corresponding maximum amplitude is described by the unknown variable, *x*.

We propose the law of maximum uncertainty, which is based on two principles, the principle of entropy and the mirror principle, in order to represent this information regarding the unknown variable, *x*. The main difference between these principles is the fact that, in some decision-making problems, these intervals can overlap and in others they do not.

We validate the proposed law in different experiments in the literature that reported violations to the laws of classical probabilistic theory and logic.

### 6.1. Principle of Entropy

For the case in which both intervals do not overlap, $I_{y}\cap I_{\neg y}=\emptyset$, the values of the waves that are closest to the equal distribution are chosen. By doing so, the uncertainty is maximised and the information about the probability wave is not lost.
**Principle of Maximum Entropy:** states that the probability distribution which best represents the current state of knowledge is the one with largest entropy (see [36,37,38]). For the case of a binary random variable, the highest entropy corresponds to an equal distribution,
(59)$$H=-p\left(y\right)\log_{2}\left(p\left(y\right)\right)-p\left(\neg y\right)\log_{2}\left(p\left(\neg y\right)\right)=-log_{2}\left(0.5\right)=1\;bit.$$

The value that best represents the state of knowledge is the one with the largest entropy, which is when θ=0. This results in the subjective probabilities:For p(y)≤p(¬y),
(60)$$p_{q}\left(y\right)=p\left(y\right)+2\sqrt{p(y,x)p(y,\neg x)}\;\;\;\mathrm{and}$$
$$p_{q}\left(\neg y\right)=1-p_{q}\left(y\right).$$For p(¬y)≤p(y),
(61)$$p{\left(\neg y\right)}_{q}=p\left(\neg y\right)+2\sqrt{p(\neg y,x)p(\neg y,\neg x)}\;\;\;\mathrm{and}$$
$$p_{q}\left(y\right)=1-p_{q}\left(\neg y\right).$$For p(y)=p(¬y),
(62)$$p_{q}\left(y\right)=2\sqrt{p(\neg y,x)p(\neg y,\neg x)}\approx p\left(\neg y\right)\;\;\;\mathrm{and}$$
$$p_{q}\left(\neg y\right)=1-p_{q}\left(y\right).$$

The probability waves in Figure 3c, which represent the Bayesian network in Figure 1, since $I_{y}\cap I_{\neg y}=\emptyset$, then they can be described by the interval,
$$p\left(y\right)=0.12,\;\;I_{y}=\left[0.01,0.23\right],$$
where the maximum entropy is given by $p_{q}\left(y\right)=0.23$, and
$$p\left(\neg y\right)=0.88,\;\;I_{\neg y}=\left[0.77,0.99\right],$$
where the maximum entropy is given by $p_{q}\left(\neg y\right)=0.77$.

### 6.2. Mirror Principle

For the case in which both intervals overlap, $I_{y}\cap I_{\neg y}\neq \emptyset$, then, an equal distribution maximises the uncertainty, but loses the information about the probability wave. To avoid this loss, we do not change the entropy of the system. We use only the positive interference as defined by the law of balance.

When the intervals overlap the positive interference is approximately the size of smaller probability value, since the arithmetic and geometric means approach each other (see Equation (Equation 46)). We maximise the uncertainty by mirroring the “probability values”.

In this sense, the value that best represents the state of knowledge is given by:For the case p(y)≤p(¬y), we define
(63)$$p_{q}\left(\neg y\right)=2\sqrt{p(y,x)p(y,\neg x)}\approx p\left(y\right)\;\;\;\mathrm{and}\;\;\;p_{q}\left(y\right)=1-p_{q}\left(\neg y\right).$$For the case p(¬y)≤p(y), we define
(64)$$p_{q}\left(y\right)=2\sqrt{p(\neg y,x)p(\neg y,\neg x)}\approx p\left(\neg y\right)\;\;\;\mathrm{and}\;\;\;p_{q}\left(\neg y\right)=1-p_{q}\left(y\right).$$

## 7. Empirical Validation

In this section, we validate the law of balance in psychological experiments from the literature, namely the prisoner’s dilemma game and the two stage gamble game. These experiments report human decisions that violate the Sure Thing Principle [39], consequently violating the laws of probability theory and logic.

### 7.1. Prisoner’s Dilemma Game and Probability Waves

In the prisoner’s dilemma game, there are two prisoners, prisoner *x* and prisoner *y*. They have no means of communicating with each other. Each prisoner is offered by the prosecutors a bargain: (1) testifying against the other one and betray (Defect); (2) refuse the deal and cooperate with the other one by remaining silent (Cooperate). For more information about the general problem description of the Prisoner’s Dilemma game, please consult Moreira and Wichert [1].

In the experimental setting of the Prisoner’s Dilemma game proposed in Shafir and Tversky [40], participants were presented with the payoff matrix of the game and they played a set of one-shot prisoner dilemma games each against a different opponent. During these one-shot games, participants were informed that they had randomly been selected to a bonus group, which consisted in having the information about the opponent’s strategy. Participants were able to use this information in their strategies. This means that three conditions were tested in order to verify if there were violations to the Sure Thing Principle:Participant was informed that the opponent chose to *defect*, ¬x.Participant was informed that the opponent chose to *cooperate*, *x*.Participant was not informed of the opponents choice.

This allows us to compute the following information:The probability that the prisoner *y* defects given *x* defects, p(¬y|¬x).The probability that the prisoner *y* defects given *x* cooperates, p(¬y|x).The probability that the prisoner *y* defects given there is no information present about knowing if prisoner *x* cooperates or defects. This can be expressed by
(65)p(¬y)=p(¬y,x)+p(¬y,¬x)=p(¬y|x)p(x)+p(¬y|¬x)p(¬x).
and is represented by a Bayesian network (see Figure 1) that indicates the influence between events *x* and *y*.

In Table 1, we summarise the results of several experiments of the literature concerned with the prisoner’s dilemma experiment, which correspond to slight variations to the one conducted in Shafir and Tversky [40], using different payoff matrices. In Table 1, p(¬y|¬x) corresponds to all participants who chose to defect given that they were informed that the opponent also chose to defect, normalised by the total of participants who played this condition; p(¬y|x) corresponds to all participants who chose to defect given that they were informed that the opponent chose to cooperate, normalised by the total of participants who played this condition; p(¬y) corresponds to all participants who chose to defect given that no information about the opponent’s strategy was given, normalised by the total number of participants in this condition. The last row of Table 1 labelled *Average* is simply the average of all the results reported in Table 1 as it is presented in the work of Pothos and Busemeyer [19]. The column *Sample size* corresponds to the number of participants used in each experiment.

These findings mainly suggest that, under uncertainty, a decision-maker tends to become more cooperative, and for that reason, they will attempt a more cooperative strategy.

### 7.2. Two Stage Gambling Game

In the two stage gambling game the participants were asked to play two gambles in a row where they had equal chance of winning $200 or losing $100 [1]. In the original experiment conducted by Tversky and Shafir [27], the authors used a within-subjects design where the participants were told the following:
*Imagine that you have just played a game of chance that gave you a 50% chance to win $200 and a 50% chance to lose $100. Imagine that you have already made such a bet. If you won this bet and were up $200, and were offered a chance to make the same bet a second time, would you take it? What if you lost the first bet and were down $100, would you make the same bet again? What if you do not yet know whether you won or lost the first bet and so do not yet know whether you are up or in debt, would you go ahead and make the same bet a second time?*.

Three experimental settings were tested:The participant was informed that he *lost* the first gamble, ¬x, and was asked if he wanted to play the second gamble *y*.The participant was informed that he *won* the first gamble, *x*, and was asked if he wanted to play the second gamble *y*.The participant was not informed about the outcome of the first gamble, and was asked if he wanted to play the second gamble *y*. This would by by the law of total probability
(66)p(y)=p(y|x)p(x)+p(y|¬x)p(¬x)

In Table 2, we summarise the results of several experiments of the literature concerned with the two stage gambling game. In Table 2, p(y|¬x) corresponds to all participants who chose to play given that they were informed that they had lost the first gamble, normalised by the total of participants who played this condition; p(y|x) corresponds to all participants who chose to play given that they were informed that they had won the first gamble, normalised by the total of participants who played this condition; p(y) corresponds to all participants who chose to play given that no information about the first gamble was given, normalised by the total number of participants in this condition. The last row of Table 2 labelled *Average* is simply the average of all the results reported in Table 2. The column *Sample size* corresponds to the number of participants used in each experiment.

These findings mainly suggest that, under uncertainty, a decision-maker tends to become more risk-averse, and for that reason, they tend to not bet on the second gamble, leading to a violation of the Sure Thing Principle.

### 7.3. Probability Waves in the Prisoner’s Dilemma and the Two Stage Gambling Game

Using the values of Table 1 and Table 2, we can determine the probability waves $p(\neg y,\theta_{\neg })$, p(y,θ) as indicated in Figure 4.

Table 3 summarises the intervals that describe the probability waves, the resulting probabilities, $p_{q}\left(\neg \right)$, that are based on the law of maximum uncertainty, the subjective probability, $p_{sub}\left(\neg y\right)$, and the classical probability values, p(¬y).

We compared the results that are based on probability waves and the law of maximal uncertainty with previous works in the literature that deal with predictive quantum-like models for decision making, see Table 4. The dynamic heuristic as described by Moreira and Wichert [1] used quantum-like Bayesian networks. Its parameters are determined by examples from a domain. On the other hand in the Quantum Prospect Decision Theory [41], the values do not need to be adapted to a domain and the quantum interference term is determined by the Interference Quarter Law. This means that the quantum interference term of total probability is simply fixed to a value equal to 0.25.

The results presented in Table 4 show that the dynamic heuristic (DH) and the law of maximum uncertainty (MU) are similar, however the dynamic heuristic, originally proposed in Moreira and Wichert [1], is the result of a domain specific learning function. In other words, this function that is used to compute the quantum interference terms is learned to the specific problem of disjunction effects. The law of maximum uncertainty, on the other hand, is able to address disjunction errors in a more generalised way. The dynamic heuristic performs slightly better than the law of maximum uncertainty, but the first one is the result of a learned domain specific function, while the latter is derived from the hypothesis that quantum interference waves should be balanced, rather than simply normalised with Bayes rule.

Additionally, with respect to the classical counterpart, quantum-like models offer advantages when modelling paradoxical decisions in Bayesian networks. In the study of Moreira and Wichert [42], the authors demonstrated that in terms of parameters, a classical Bayesian network would require more random variables (and consequently more parameters) to reproduce the paradoxical results reported in the several experiments of the literature showing disjunction effects than its quantum-like counterpart. This suggests that quantum-like models might offer advantages, not only in terms of a cognitive perspective, but also computationally.

## 8. Quantum-like Bayesian Network

To apply the law of balance to Quantum-Like Bayesian networks, our next step is to generalise the law of balance rule during probabilistic inference from one unknown variable, *x*, to several, $x_{1},x_{2},x_{3},\cdots ,x_{N}$.

### 8.1. Generalisation

For the binary event *x* and non binary event *y*, we have that
(67)$$p\left(x\right)+p\left(\neg x\right)=1,\;\;\sum_{i=1}^{M}p\left(y_{i}\right)=1,$$
and the law of total probability is represented by
(68)$$p\left(y\right)=\sum_{i=1}^{M}p\left(y,x_{i}\right)=\sum_{i=1}^{M}p\left(y|x_{i}\right)p\left(x_{i}\right),$$
(69)$$p\left(\neg y\right)=\sum_{i=1}^{M}p\left(\neg y,x_{i}\right)=\sum_{i=1}^{M}p\left(\neg y|x_{i}\right)p\left(x_{i}\right).$$

The intensity wave is defined as
(70)$$I\left(y,\theta_{1},\theta_{2},\cdots ,\theta_{M}\right)={\left|\sum_{i=1}^{M}A\left(y,x_{i}\right)\right|}^{2}$$
(71)$$I\left(y,\theta_{1},\theta_{2},\cdots ,\theta_{M}\right)=p\left(y\right)+2\sum_{i=1}^{M-1}\sum_{j=i+1}^{M}\sqrt{p\left(y,x_{i}\right)p\left(y,x_{j}\right)}\cos\left(\theta_{i}-\theta_{j}\right),$$
(72)$$I\left(\neg y,\theta_{\neg 1},\cdots ,\theta_{\neg M}\right)=p\left(\neg y\right)+2\sum_{i=1}^{M-1}\sum_{j=i+1}^{M}\sqrt{p\left(\neg y,x_{i}\right)p\left(\neg y,x_{j}\right)}\cos\left(\theta_{\neg i}-\theta_{\neg j}\right),$$
usually there are interference terms during probabilistic inferences in quantum-like Bayesian networks, therefore the intensity waves do not follow the rules of classical probability theory,
(73)$$I\left(y,\theta_{1},\theta_{2},\cdots ,\theta_{M}\right)+I\left(\neg y,\theta_{\neg 1},\theta_{\neg 2},\cdots ,\theta_{\neg M}\right)\neq 1.$$
The intensity waves, $I(z,\theta_{1},\theta_{2},\cdots ,\theta_{M})$, is always non negative, since the norm is non-negative,
(74)$$0\leq I\left(y.\theta_{1},\theta_{2},\cdots ,\theta_{M}\right)={\left|\sum_{i=1}^{M}a\left(y,x_{i},\theta_{i}\right)\right|}^{2}={∥\sum_{i=1}^{M}a\left(y,x_{i},\theta_{i}\right)∥}^{2}.$$

### 8.2. Probability Waves Sum to One according by the Law of Balance

For *M* unknown events, the interference from the intensity waves cancel each other, if they satisfy the condition,
(75)$$I\left(y,\theta_{1},\theta_{2},\cdots ,\theta_{M}\right)+I\left(\neg y,\theta_{\neg 1},\theta_{\neg 2},\cdots ,\theta_{\neg M}\right)=1$$
if
$$\sum_{i=1}^{M-1}\sum_{j=i+1}^{M}\sqrt{p\left(y,x_{i}\right)p\left(y,x_{j}\right)}\cos\left(\theta_{i}-\theta_{j}\right)=$$
(76)$$-\sum_{i=1}^{M-1}\sum_{j=i+1}^{M}\sqrt{p\left(\neg y,x_{i}\right)p\left(\neg y,x_{j}\right)}\cos\left(\theta_{\neg i}-\theta_{\neg j}\right).$$

The relationship between different combinations of phases $\theta_{\neg i}-\theta_{\neg j}$ cannot be simplified. It is not possible to define a $\theta_{\neg }$ as before. If we decompose Equation (Equation 76) for each $\theta_{i}$ and $\theta_{j}$ we get
(77)$$\lambda =\frac{M!}{(M-2)!2!}=\frac{M(M-1)}{2}$$
parameters and the sub-equations
(78)$$\sqrt{p\left(y,x_{i}\right)p\left(y,x_{j}\right)}\cos\left(\theta_{i}-\theta_{j}\right)=-\sqrt{p\left(\neg y,x_{i}\right)p\left(\neg y,x_{j}\right)}\cos\left(\theta_{\neg i}-\theta_{\neg j}\right)$$
representing λ subsystems that can be solved independently as before.

### 8.3. Probability Waves are Smaller Equal One only after Normalisation

The general formula for interference effects in quantum-like Bayesian networks is given by
(79)$$I\left(y,\theta_{1},\cdots ,\theta_{M}\right)=\sum_{i=1}^{M}p\left(y,x_{i}\right)+2\sum_{i=1}^{M-1}\sum_{j=i+1}^{M}\sqrt{p\left(y,x_{i}\right)p\left(y,x_{j}\right)}\cos\left(\theta_{i}-\theta_{j}\right).$$
By the inequality of arithmetic and geometric means, if follows that
(80)$$\sqrt{p\left(y,x_{i}\right)p\left(y,x_{j}\right)}\leq \frac{p\left(y,x_{i}\right)+p\left(y,\neg x_{j}\right)}{2},$$
(81)$$2\sqrt{p\left(y,x_{i}\right)p\left(y,\neg x_{j}\right)}\leq p\left(y,x_{i}\right)+p\left(y,\neg x_{j}\right).$$

There are M/2 pairs $p\left(y,x_{i}\right)+p\left(y,\neg x_{j}\right)$, since the summation goes over *M*. There are $\lambda =\frac{M(M-1)}{2}$ interference sub-equations $2\sqrt{p\left(y,x_{i}\right)p\left(y,\neg x_{j}\right)}$. It means that there are M−1 more interference sub-equations. For *M* the intensity wave becomes a probability wave if we normalise the interference part by dividing through M−1. Note for M=2, we do not need to normalise since the value is one. The intensity wave is a probability wave by using the law of balance and by normalising the interference part by dividing through M−1. This results in
(82)$$p\left(y,\theta_{1},\cdots ,\theta_{M}\right)=\sum_{i=1}^{M}p\left(y,x_{i}\right)+\frac{2}{M-1}\sum_{i=1}^{M-1}\sum_{j=i+1}^{M}\sqrt{p\left(y,x_{i}\right)p\left(y,x_{j}\right)}\cos\left(\theta_{i}-\theta_{j}\right)$$
(83)$$p\left(\neg y,\theta_{1},\cdots ,\theta_{M}\right)=\sum_{i=1}^{M}p\left(\neg y,x_{i}\right)+\frac{2}{M-1}\sum_{i=1}^{M-1}\sum_{j=i+1}^{M}\sqrt{p\left(\neg y,x_{i}\right)p\left(\neg y,x_{j}\right)}\cos\left(\theta_{i}-\theta_{j}\right)$$

The interference, however, is not symmetric, since it is composed out of a sum of sub-equations.
$$Interference_{max}\neq -Interference_{max}$$

### 8.4. Example of Estimation of Balanced Phases

In the following example, we present the probability wave $p(y,\theta_{1},\theta_{2},\theta_{3})$,
$$p\left(y,\theta_{1},\theta_{2},\theta_{3}\right)=p\left(y,x_{1}\right)+p\left(y,x_{2}\right)+p\left(y,x_{3}\right)+\sqrt{p\left(y,x_{1}\right)p\left(y,x_{2}\right)}\cos\left(\theta_{1}-\theta_{2}\right)+$$
(84)$$+\sqrt{p\left(y,x_{1}\right)p\left(y,x_{3}\right)}\cos\left(\theta_{1}-\theta_{3}\right)+\sqrt{p\left(y,x_{2}\right)p\left(y,x_{3}\right)}\cos\left(\theta_{2}-\theta_{3}\right)$$
with
(85)0≤p(y,θ)≤1.
The relation between different combinations of phases $\theta_{i}-\theta_{j}$ cannot be simplified. It is not possible to define a θ as before since two out of three permutations exist. Having two different phases there is only one combination and we can project the two dimensional function onto one dimension function (see Figure 5a). With three different phases, a three dimensional function cannot be projected onto one dimension.

If we assume the following values,
$$p\left(x_{1}\right)=0.2,\;p\left(x_{2}\right)=0.5,\;p\left(x_{3}\right)=0.3$$
and
$$p\left(y|x_{1}\right)=0.13,\;p\left(y|x_{2}\right)=0.33,p\left(y|x_{3}\right)=0.23,$$
in Figure 5b–d, we assume that each of the three phases of $p(y,\theta_{1},\theta_{2},\theta_{3})$ is zero. This results in three different plots which approximate the three dimensional function by three projections onto two dimensions.

Again, we determined $\pm Interference_{max}$ of the three or more dimensional waves numerically. We chose the balanced phases according to Equation (Equation 78) and determined the minima and maxima of the three dimensional Equation (Equation 84) wave numerically by a numerical computing environment. We used *Wolfram Mathematica* using the build in functions Maximize and Minimize, the source code will be publicly available in Github: https://github.com/catarina-moreira/QuLBIT.

When $I_{y}\cap I_{\neg y}=\emptyset$, then we obtain
$$p\left(y\right)=0.26,\;\;\;\;I_{y}=\left[0.13,0.48\right],\;\;\;\;p_{q}\left(y\right)=0.48,$$
$$p\left(\neg y\right)=0.74,\;\;\;\;I_{\neg y}=\left[0.52,0.87\right],\;\;\;\;p_{q}\left(\neg y\right)=0.52.$$

### 8.5. Example of Application in the Burglar / Alarm Bayesian Network

For the Bayesian Network represented in Figure 2, for unknown variables like $x_{3}$, using the ignorance rule we get Equation (Equation 86)
$$p\left(x_{4},\theta_{i},\theta_{ii}|x_{1},x_{2}\right)=\alpha (p\left(x_{2}\right)p\left(x_{4}|x_{1}\right)(p\left(x_{1}|x_{2},x_{3}\right)p\left(x_{3}\right)+p\left(x_{1}|x_{2},\neg x_{3}\right)p\left(\neg x_{3}\right)+$$
(86)$$+2\sqrt{p\left(x_{1}|x_{2},x_{3}\right)p\left(x_{3}\right)p\left(x_{1}|x_{2},\neg x_{3}\right)p\left(\neg x_{3}\right)}\cos\left(\theta_{i}-\theta_{ii}\right))$$
and Equation (Equation 87).
$$p\left(\neg x_{4},\theta_{\neg i},\theta_{\neg ii}|x_{1},x_{2}\right)=\alpha (p\left(x_{2}\right)p\left(\neg x_{4}|x_{1}\right)(p\left(x_{1}|x_{2},x_{3}\right)p\left(x_{3}\right)+p\left(x_{1}|x_{2},\neg x_{3}\right)p\left(\neg x_{3}\right)+$$
(87)$$+2\left(\sqrt{p\left(x_{1}|x_{2},x_{3}\right)p\left(x_{3}\right)p\left(x_{1}|x_{2},\neg x_{3}\right)p\left(\neg x_{3}\right)}\cos\left(\theta_{\neg i}-\theta_{\neg ii}\right)\right)$$

We use the roman notation i,ii,iii,vi,⋯ for the index of the phase to distinguish between the index of variables. Two solutions exist:For the constraint,
(88)$$\frac{p(\neg x_{4}|x_{1})}{p(x_{4}|x_{1})}\leq 1,$$
we get
(89)$$\theta_{i}-\theta_{ii}=\cos^{-1}\left(-\frac{p(\neg x_{4}|x_{1})}{p(x_{4}|x_{1})}\cos\left(\theta_{\neg i}-\theta_{\neg ii}\right)\right)$$For the constraint,
(90)$$\frac{p(x_{4}|x_{1})}{p(\neg x_{4}|x_{1})}\leq 1,$$
we get
(91)$$\theta_{\neg i}-\theta_{\neg ii}=\cos^{-1}\left(-\frac{p(x_{4}|x_{1})}{p(\neg x_{4}|x_{1})}\cos\left(\theta_{i}-\theta_{ii}\right)\right)$$

Using the parameters of the Bayesian network in Figure 2, we get
(92)$$p\left(x_{4}|x_{1},x_{2}\right)=0.9,\;\;p\left(\neg x_{4}|x_{1},x_{2}\right)=0.1.$$
Since the priors are small, $Burglary(=x_{2}=0.001$), $Earthquake(=x_{3}=0.002$), the quantum interference effects will not become noticeable (see Figure 6).

For $I_{y}\cap I_{\neg y}=\emptyset$, we obtain
$$p\left(x_{4}|x_{1}\right)=0.9,\;\;\;\;I_{y}=\left[0.899992,0.900008\right],\;\;\;\;p_{q}\left(x_{4}|x_{1}\right)=0.899992$$
$$p\left(\neg x_{4}|x_{1}\right)=0.1,\;\;\;\;I_{\neg y}=\left[0.099992,0.100008\right],\;\;\;\;p_{q}\left(\neg x_{4}|x_{1}\right)=0.100008.$$

By increasing the parameter values of $Burglary(=x_{2}=0.5$) and $Earthquake(=x_{3}=0.2$), the quantum interference effects become noticeable. Again, we estimated numerically with the new increased parameters. With $I_{y}\cap I_{\neg y}=\emptyset$, we obtained
$$p\left(x_{4}|x_{1}\right)=0.9,\;\;\;\;I_{y}=\left[0.862201,0.937799\right],\;\;\;\;p_{q}\left(x_{4}|x_{1}\right)=0.862201$$
$$p\left(\neg x_{4}|x_{1}\right)=0.1,\;\;\;\;I_{\neg y}=\left[0.062201,0.137799\right],\;\;\;\;p_{q}\left(\neg x_{4}|x_{1}\right)=0.137799.$$

In this case, the interference part becomes noticeable (see Figure 7). However, the increase does not have any effect on the classical probabilities since the values are only dependent on the value $p(x_{4}|x_{1})$.
(93)$$p\left(x_{4}|x_{1},x_{2}\right)=\frac{p(x_{4}|x_{1})}{p\left(x_{4}|x_{1}\right)+p\left(\neg x_{4}|x_{1}\right)}.$$

For unknown variables $x_{3}$, $x_{1}$ we get the four dimensional probability wave we get the Equation (Equation 94). One can clearly see that the interference value diminish since six we multiply for each interference part seven probability values.
$$p\left(x_{4},\theta_{i},\theta_{iii}.\theta_{i},\theta_{iv}|x_{2}\right)=\alpha (p\left(x_{2}\right)(p\left(x_{4}|x_{1}\right)p\left(x_{1}|x_{2},x_{3}\right)p\left(x_{3}\right)+p\left(x_{4}|x_{1}\right)p\left(x_{1}|x_{2},\neg x_{3}\right)p\left(\neg x_{3}\right)+$$
$$+p\left(x_{4}|\neg x_{1}\right)p\left(\neg x_{1}|x_{2},x_{3}\right)p\left(x_{3}\right)+p\left(x_{4}|\neg x_{1}\right)p\left(\neg x_{1}|x_{2},\neg x_{3}\right)p\left(\neg x_{3}\right))+$$
$$+2/3\sqrt{p\left(x_{4}|x_{1}\right)p\left(x_{1}|x_{2},x_{3}\right)p\left(x_{3}\right)p\left(x_{4}|x_{1}\right)p\left(x_{1}|x_{2},\neg x_{3}\right)p\left(\neg x_{3}\right)}\cos\left(\theta_{i}-\theta_{ii}\right)+$$
$$+2/3\sqrt{p\left(x_{4}|x_{1}\right)p\left(x_{1}|x_{2},x_{3}\right)p\left(x_{3}\right)p\left(x_{4}|\neg x_{1}\right)p\left(\neg x_{1}|x_{2},x_{3}\right)p\left(x_{3}\right)}\cos\left(\theta_{i}-\theta_{iii}\right)+$$
$$+2/3\sqrt{p\left(x_{4}|x_{1}\right)p\left(x_{1}|x_{2},x_{3}\right)p\left(x_{3}\right)p\left(x_{4}|\neg x_{1}\right)p\left(\neg x_{1}|x_{2},\neg x_{3}\right)p\left(\neg x_{3}\right)}\cos\left(\theta_{i}-\theta_{iv}\right)+$$
$$+2/3\sqrt{p\left(x_{4}|x_{1}\right)p\left(x_{1}|x_{2},\neg x_{3}\right)p\left(\neg x_{3}\right)p\left(x_{4}|\neg x_{1}\right)p\left(\neg x_{1}|x_{2},x_{3}\right)p\left(x_{3}\right)}\cos\left(\theta_{ii}-\theta_{iii}\right)+$$
$$+2/3\sqrt{p\left(x_{4}|x_{1}\right)p\left(x_{1}|x_{2},\neg x_{3}\right)p\left(\neg x_{3}\right)p\left(x_{4}|\neg x_{1}\right)p\left(\neg x_{1}|x_{2},\neg x_{3}\right)p\left(\neg x_{3}\right)}\cos\left(\theta_{ii}-\theta_{iv}\right)+$$
(94)$$+2/3\sqrt{p\left(x_{4}|\neg x_{1}\right)p\left(\neg x_{1}|x_{2},x_{3}\right)p\left(x_{3}\right)p\left(x_{4}|\neg x_{1}\right)p\left(\neg x_{1}|x_{2},\neg x_{3}\right)p\left(\neg x_{3}\right)}\cos\left(\theta_{iii}-\theta_{iv}\right)).$$

With the normalisation factor α,
(95)$$\alpha =\frac{1}{p(x_{2})}=\frac{1}{p\left(X_{1},x_{2},X_{3},x_{4}\right)+p\left(X_{1},x_{2},X_{3},\neg x_{4}\right)}.$$
we get six sub-equations of which each has two solutions with constraints that are defined by three different equations and their symmetrical counterparts
1st Equation:
(96)$$\theta_{i}-\theta_{ii}=\cos^{-1}\left(-\frac{p(\neg x_{4}|x_{1})}{p(x_{4}|x_{1})}\cos\left(\theta_{\neg i}-\theta_{\neg ii}\right)\right).$$2nd Equation:
(97)$$\theta_{i}-\theta_{iii}=\cos^{-1}\left(-\frac{p\left(\neg x_{4}|x_{1}\right)p\left(\neg x_{4}|\neg x_{1}\right)}{p\left(x_{4}|x_{1}\right)p\left(x_{4}|\neg x_{1}\right)}\cos\left(\theta_{\neg i}-\theta_{\neg iii}\right)\right)$$
which is also valid for $\theta_{i}-\theta_{iv}$, $\theta_{ii}-\theta_{iii}$ and $\theta_{ii}-\theta_{iv}$.3rd Equation:
(98)$$\theta_{iii}-\theta_{iv}=\cos^{-1}\left(-\frac{p(\neg x_{4}|\neg x_{1})}{p(x_{4}|\neg x_{1})}\cos\left(\theta_{\neg iii}-\theta_{\neg iv}\right)\right).$$The balanced phases are computed as before.
Using the parameters of the Bayesian network in Figure 2, we can determine the balanced wave
$$p(x_{4},\theta_{i},\theta_{ii},\theta_{iii},\theta_{iv}|x_{2}).$$
In order to do this, we have to determine the minima or maxima of the four dimensional wave described by Equation (Equation 94) numerically. We used *Mathematica* using the build in functions Maximize and Minimize, the source code will be publicly available in Github: https://github.com/catarina-moreira/QuLBIT. Since the parameter values are small $Burglary(=x_{2}=0.001$), $Earthquake(=x_{3}=0.002$), the interference part becomes nearly not noticeable as before. With, $I_{y}\cap I_{\neg y}=\emptyset$, we obtain
$$p\left(x_{4}|x_{1}\right)=0.857072,\;\;\;\;I_{y}=\left[0.857047,0.857094\right],\;\;\;\;p_{q}\left(x_{4}|x_{1}\right)=0.857047,$$
$$p\left(\neg x_{4}|x_{1}\right)=0.142928,\;\;\;\;I_{\neg y}=\left[0.14288,0.142979\right],\;\;\;\;p_{q}\left(\neg x_{4}|x_{1}\right)=0.142979.$$

By increasing the parameter values to $Burglary(=x_{2}=0.5$), $Earthquake(=x_{3}=0.2$), the interference part becomes noticeable. We estimate numerically with the increased parameters with, $I_{y}\cap I_{\neg y}=\emptyset$, we obtain
$$p\left(x_{4}|x_{1}\right)=0.857157,\;\;\;\;I_{y}=\left[0.80666,0.899603\right],\;\;\;\;p_{q}\left(x_{4}|x_{1}\right)=0.80666$$
$$p\left(\neg x_{4}|x_{1}\right)=0.142843,\;\;\;\;I_{\neg y}=\left[0.108414,0.191128\right],\;\;\;\;p_{q}\left(\neg x_{4}|x_{1}\right)=0.191128.$$

It seems that, under the law of balance, the quantum interference tends to diminish with the complexity of the decision scenario as indicated with the examples of balanced quantum-like Bayesian network.

## 9. Interpretation

According to most physics textbooks, the existence of the wave function and its collapse is only present in the microscopic world and is not present in the macroscopic world. However, there has been an increasing amount of scientific studies indicating that this is not true and that wave functions are indeed present in the macroscopic world (see Vedral [43]). Additionally, experiments in physics state that the size of atoms does not matter and that a very large number of atoms actually be entangled (see Amico et al. [44], Ghosh et al. [45]).

Clues from psychology indicate that human cognition is based on quantum probability rather than the traditional probability theory as explained by Kolmogorov’s axioms [7,16,17,18]. It seems that under uncertainty, Kolmogorov’s axioms tell what the decision-maker should choose, while quantum probability can indicate what the decision-maker actually chooses [46]. This could tell us that, under a cognitive point of view, a wave function can indeed be present at the macro scale of our daily life. This implies a unified explanation of human cognition under the paradigm of quantum cognition and quantum interference.

A unified explanation of human interference using quantum probability theory and classical theory was for the first time proposed in Trueblood et al. [47]. The authors propose a hierarchy of mental representations ranging from quantum-like to classical. This approach allows the combination of both Bayesian and non-Bayesian influences in cognition, where classical representations provide a better account of data as individuals gain familiarity, and quantum-like representations can provide novel predictions and novel insights about the relevant psychological principles involved during the decision process.

Motivated by this model, we propose the distinction between *unknown*, which can be seen as a *truth* value, and *ignorance* as the *lack of knowledge* or as being unaware. An event can either be *true*, *false* or *unknown*. The proposed balanced quantum-like approach can be integrated in this view in the following way:For *unknown* events, the classical law of total probability is applied;For events of which we are unaware, we apply quantum-like models in which the phase information is related to *ignorance*. We determine the possible values of the wave using the law of maximum entropy of quantum-like systems.

Note that, an *unknown* event is not known to the decision-maker, because he does not have enough information. *Ignorance* means that the decision-maker cannot obtain this information, so ignorance is not a truth value at all and the decision-maker does not know at what the value of the event is. This relationship is analogous to the relation between pseudo randomness and true randomness. Pseudo randomness appears to the decision-maker as being totally random due to his lack of information. True quantum randomness corresponds to ignorance.

This distinction between unknown and ignorance can also be explained by the interference in the two-slit experiment.
In the two-slit experiment, with the electron detectors showing which slit the electron goes through, the electron behaves as a particle. Assuming that the information about the detectors is unknown to us, we apply the law of total probability.When the detectors are removed, the electron is unobserved and is represented as a wave. In this case, we apply the quantum-like law of total probability.

Under this context, ignorance corresponds to a prediction in the future.

## 10. Conclusions

This work is motivated by empirical findings from cognitive psychology, which indicate that, in scenarios under high levels of uncertainty, most people tend to make decisions that violate the laws of classical probability and logic. It seems that normative models, like Bayesian inference and the expected utility theory, tend to compute what people *should* choose, instead of computing what they actually choose [46]. In order to address this issue, many models have been proposed in the literature, which are based on quantum probability theory.

In this work, we explored the relationship between the empirical findings from cognitive psychology and quantum probability amplitudes. More specifically, we make use of the notion of *intensity waves* as the interference effects that result in the double slit experiment, when there are no detectors in the slits. We then investigated the relationship between the phase of these intensity waves and the resulting subjective probabilities that were found in the different experiments reported in the literature, showing paradoxical human decisions. We found that there is indeed a meaningful relationship between the phase and the resulting subjective probability, which is the result of a different, and novel, normalisation method that is summarised into two laws:The Law of Balance: a novel mathematical formalism for quantum-like probabilistic inferences that enables the cancellation of the quantum interference terms upon the application of Bayes normalisation factor. This way, the amplitudes of the probability waves become balanced.The Law of Maximum Uncertainty: which states that in order to predict disjunction effects, one should choose the amplitude of the wave that contains most the maximum uncertainty, or the the maximum information.

These laws were used in the formalism of quantum-like Bayesian networks, in a model that we define as the *balanced quantum-like Bayesian network*, in order to model the disjunction effects and to represent uncertainty. We validated the proposed balanced quantum-like Bayesian network in the different experiments reported in the literature, mainly disjunction effects under the Prisoner’s Dilemma game and the Two-Stage gambling game. Results showed that the proposed quantum-like Bayesian network could predict many of these disjunction effects.

Although we cannot test the proposed approach in more complex decision problems due to the current nonexistence of data, our analysis indicated that the quantum interference seem to diminish with the complexity of the decision scenario as indicated before with the examples of balanced quantum-like Bayesian networks. There are, however, some preliminary studies on real-world complex decision-scenarios of quantum-like Bayesian networks, namely trying to predict the probability of a client receiving a credit or not in a financial institution [48]. This preliminary analysis shows that, under uncertainty, the quantum-like Bayesian network could fit the data better due to quantum interference effects and was able to reproduce better the underlying credit application process of the financial institution better than the classical network. This study indicates the potential of quantum-like decision technologies, however this is still an open question and more research is needed in this direction.

## Figures and Tables

**Figure 1 entropy-22-00170-f001:**
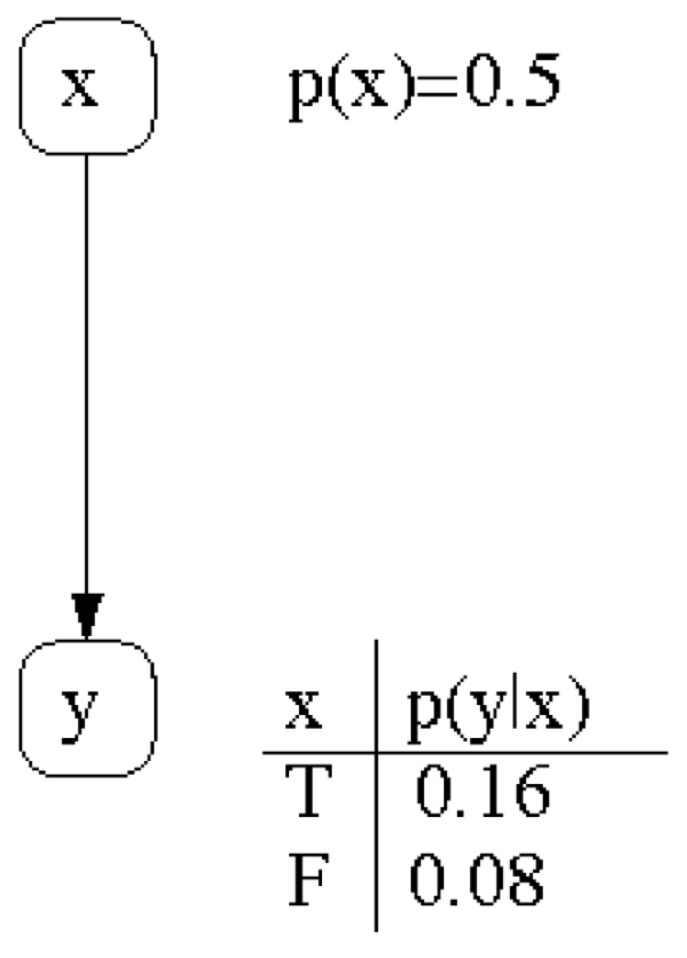
The probabilistic influence of random variables *X* on *Y* represented by a Bayesian network. Note that each node is followed by a conditional probability table that specifies the probability distribution of how node *Y* is conditioned by node *X*.

**Figure 2 entropy-22-00170-f002:**
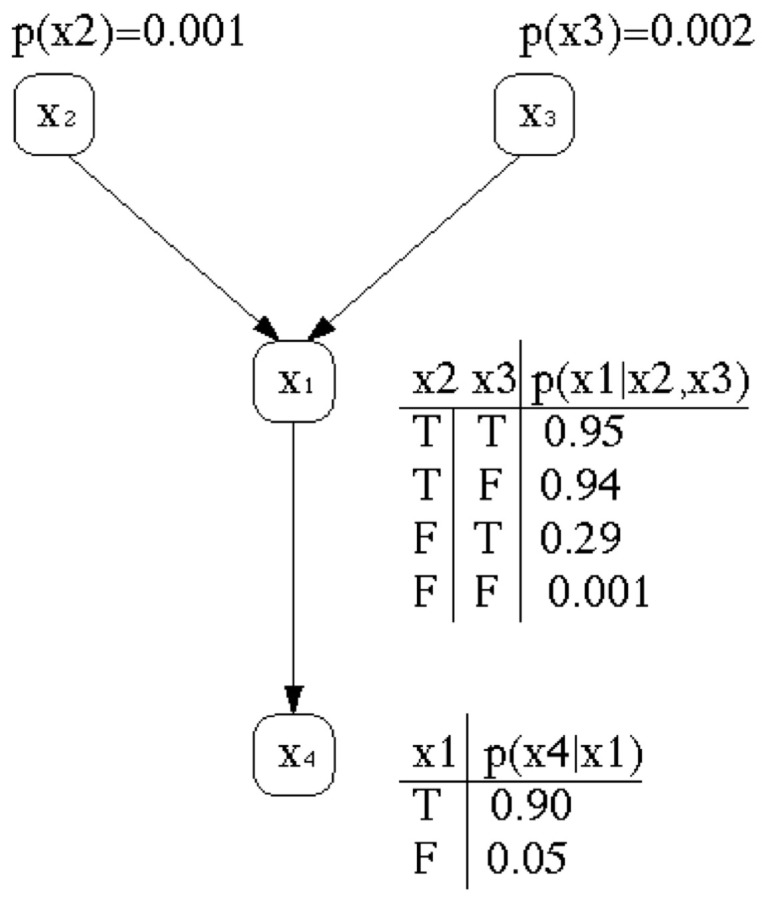
A Bayesian network representing the causal relationship between events $x_{1}$, $x_{2}$, $x_{3}$ and $x_{4}$. The four variables can be associated with causal knowledge, in our example $Burglary(=x_{2}$), $Earthquake(=x_{3}$), $Alarm(=x_{1}$) and $JohnCalls(=x_{4}$).

**Figure 3 entropy-22-00170-f003:**
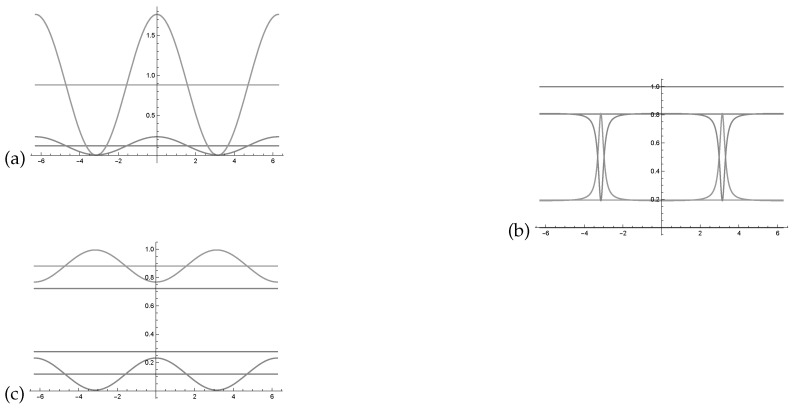
(**a**) Two intensity waves I(y,θ), $I(\neg y,\theta_{\neg })$ in relation to the phase (−2π,2π) with the parametrisation as indicated in corresponding to the values of Figure 1. Note that the two waves oscillate around p(y)=0.1950 and p(¬y)=0.8050 (the two lines). (**b**) Normalisation of the two intensity waves I(y,θ), $I(\neg y,\theta_{\neg })$. The two normalised waves do not oscillate around p(y) and p(¬y). (**c**) The resulting probability waves as determined by the law of balance, the bigger wave is replaced by the negative smaller one.

**Figure 4 entropy-22-00170-f004:**
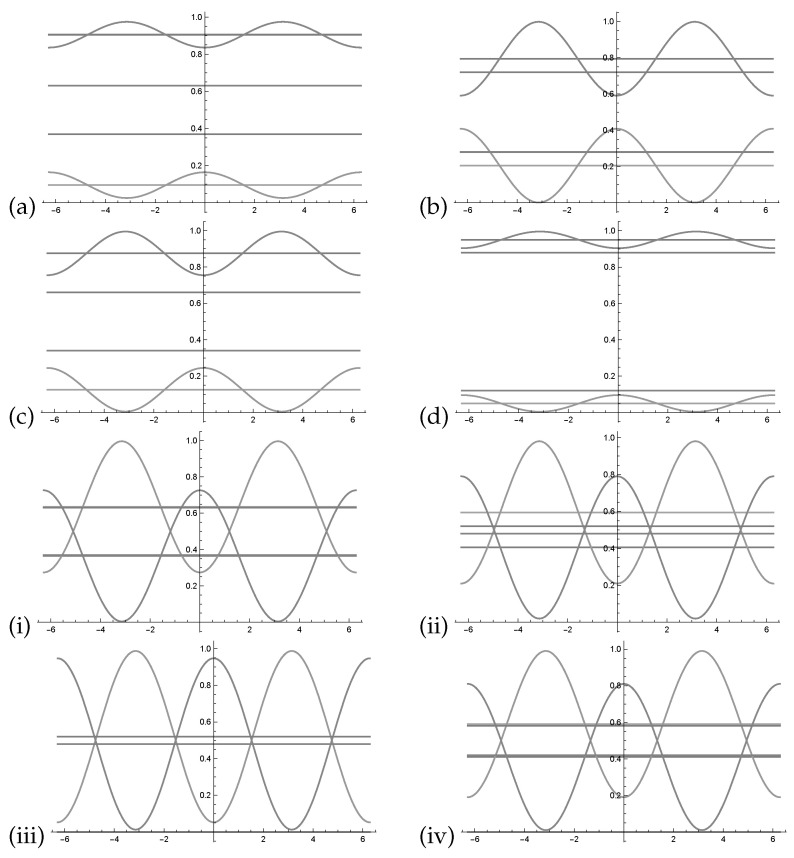
Probability waves for the experiments described in Table 1 and Table 2. In plots (**a**–**d**) the waves $p(\neg y,\theta_{\neg })$ are around p(¬y), (for (**e**) see Figure 3c). In the plots (**i**–**iv**) the waves p(y,θ) are around p(y). Additionally the values $p_{sub}\left(\neg y\right)$ and $p_{sub}\left(y\right)$ are indicated by a line. Note that the curves in the plots (**i**–**iv**) overlap.

**Figure 5 entropy-22-00170-f005:**
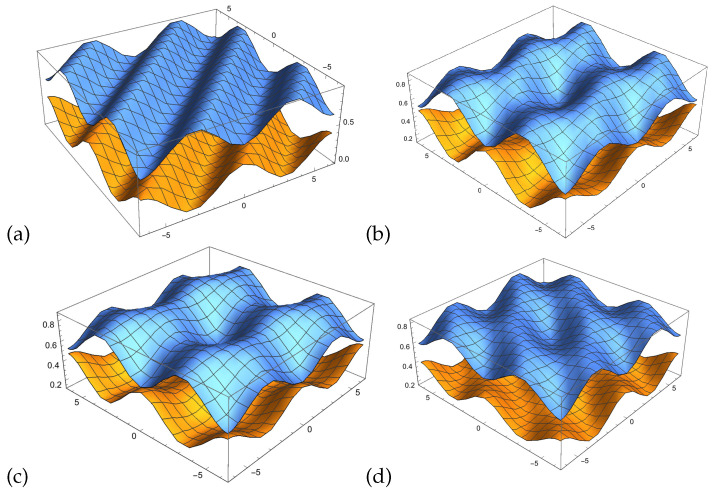
(**a**) Having two different phases there is only one combination and we can project the two dimensional function onto one dimension function. The cos function in the relation $\theta_{1}-\theta_{2}$. In (**b**–**d**) we assume that each of the three phases of $p(y,\theta_{1},\theta_{2},\theta_{3})$ and $p(\neg y,\theta_{\neg 1},\theta_{\neg 2},\theta_{\neg 3})$ is zero and get three different plots which approximate the three dimensional function by three projections onto two dimension. In (**b**) we assume $\theta_{\neg 3}=\theta_{3}=0$. In (**c**) we assume $\theta_{\neg 2}=\theta_{2}=0$. In (**d**) we assume $\theta_{\neg 1}=\theta_{1}=0$.

**Figure 6 entropy-22-00170-f006:**
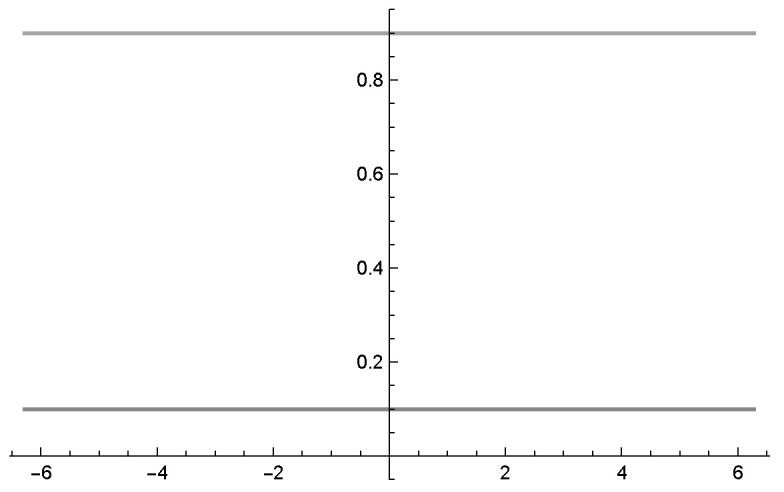
Probability waves. Since the parameter values are small $Burglary(=x_{2}=0.001$), $Earthquake(=x_{3}=0.002$), the interference part is not noticeable.

**Figure 7 entropy-22-00170-f007:**
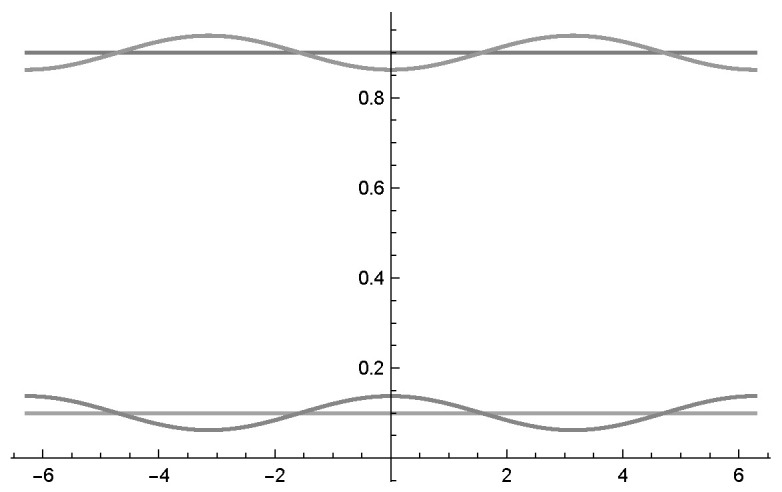
Probability waves. Since the parameter are increased $Burglary(=x_{2}=0.5$), $Earthquake(=x_{3}=0.2$), the interference part is noticeable.

**Table 1 entropy-22-00170-t001:** Experimental results obtained in four different works of the literature for the prisoner’s dilemma game. The column p(¬y|¬x) corresponds to the probability of defecting given that it is known that the other participant chose to defect. The column p(¬y|x) corresponds to the probability of defecting given that it is known that the other participant chose to cooperate. The column $p_{sub}\left(\neg y\right)$ corresponds to the subjective probability of the second participant choosing the defect action given there is no information present about knowing if prisoner *x* cooperates or defects. The column p(¬y) corresponds to the classical probability. Finally, the column *Sample Size* describes the number of participants used in each experiment of the Prisoner’s Dilemma game. ${}^{a}$ corresponds to the average results of all seven experiments reported.

Experiment	p(¬y|¬x)	p(¬y|x)	$p_{sub}\left(\neg y\right)$	p(¬y)	Sample Size
(a) Tversky and Shafir [27]	0.97	0.84	0.63	0.9050	80
(b) Li and Taplin [26] ${}^{a}$	0.82	0.77	0.72	0.7950	30
(c) Busemeyer et al. [24]	0.91	0.84	0.66	0.8750	88
(d) Hristova and Grinberg [25]	0.97	0.93	0.88	0.9500	20
(e) Average	0.92	0.85	0.72	0.8813	54

**Table 2 entropy-22-00170-t002:** Experimental results obtained in three different works of the literature indicating the probability of a player choosing to make a second gamble for the two stage gambling game. The column p(y|¬x) corresponds to the probability when the outcome of the first gamble is known to be lost. The column p(y|x) corresponds to the probability when the outcome of the first gamble is known to be win. Finally, the column $p_{sub}\left(y\right)$ corresponds to the subjective probability when the outcome of the first gamble is not known. The column p(y) corresponds to the classical probability. ${}^{a}$ corresponds to the average results of all four experiments reported.

Experiment	p(y|¬x)	p(y|x)	$p_{sub}\left(y\right)$	p(y)	Sample Size
(i) Tversky and Shafir [27]	0.58	0.69	0.37	0.6350	98
(ii) Kuhberger et al. [28] ${}^{a}$	0.47	0.72	0.48	0.5950	135
(iii) Lambdin and Burdsal [29]	0.45	0.63	0.41	0.5400	57
(iv) Average	0.50	0.68	0.42	0.5900	96

**Table 3 entropy-22-00170-t003:** Probability waves, the resulting probabilities $p_{q}$ that are based on the law of maximal uncertainty, the subjective probability and the classical probability values. Entries (a)–(e) are based on the principle of entropy and entries (i)–(iv) are based mirror principle.

**Experiment: Prisoner’s Dilemma**	$I_{\neg y}$	$p_{sub}\left(\neg y\right)$	$p_{q}\left(\neg y\right)$	p(¬y)
(a) Tversky and Shafir [27]	[0.84, 0.97]	0.63	0.84	0.91
(b) Li and Taplin [26]	[0.59, 1.00]	0.72	0.59	0.79
(c) Busemeyer et al. [24]	[0.76, 1.00]	0.66	076	0.88
(d) Hristova and Grinberg [25]	[0.90,1.00]	0.88	0.90	0.95
(e) Average	[0.77,0.99]	0.72	0.77	0.88
**Experiment: Two Stage Gamble**	$I_{y}$	$p_{sub}\left(y\right)$	$p_{q}\left(y\right)$	p(y)
(i) Tversky and Shafir [27]	[0.27, 0.98]	0.37	0.36	0.64
(ii) Kuhberger et al. [28]	[0.20, 0.98]	0.48	0.39	0.59
(iii) Lambdin and Burdsal [29]	[0.09, 0.99]	0.41	0.45	0.54
(vi) Average	[0.19, 0.99]	0.42	0.40	0.59

**Table 4 entropy-22-00170-t004:** Comparison between the Quantum Prospect Decision Theory (DT), see Yukalov and Sornette [41], the dynamic heuristic (DH), see Moreira and Wichert [1] and the law of maximal uncertainty (MU) of the balanced quantum-like model. The results of the dynamic heuristic (DH) and the law of maximal uncertainty (MU) are similar, however the the law of maximal uncertainty (MU) was not adapted to a domain.

**Experiment: Prisoner’s Dilemma**	***observed***	***PDT***	***DH***	***MU***
(a) Tversky and Shafir [27]	0.63	0.65	0.64	0.84
(b) Li and Taplin [26]	0.72	0.54	0.71	0.59
(c) Busemeyer et al. [24]	0.66	0.63	0.80	0.76
(d) Hristova and Grinberg [25]	0.88	0.70	0.90	0.90
(e) Average	0.72	0.63	0.76	0.77
**Experiment: Two-Stage Gamble**	***observed***	***PDT***	***DH***	***MU***
(i) Tversky and Shafir [27]	0.37	0.39	0.36	0.36
(ii) Kuhberger et al. [28]	0.48	0.35	0.40	0.39
(iii) Lambdin and Burdsal [29]	0.41	0.29	0.41	0.45
(iv) Average	0.42	0.34	0.39	0.40

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
