# Peer review of "Balanced Quantum-Like Bayesian Networks"

_entropy, 2020, doi:10.3390/e22020170_

Round 1
Reviewer 1 Report
In this work, the authors studied some psychology problem using quantum-like method. They proposed a Law of Balance for probabilistic inferences in quantum-like Bayesian networks. They also proposed the law of maximum entropy, to predict the paradoxes by selecting the amplitudes of the wave with the highest uncertainty. Empirical results showed the two proposed laws were able to predict experiments from cognitive psychology showing paradoxical or irrational decisions, namely in the Prisoner’s Dilemma game and the Two-Stage Gambling Game. The results are interesting and could be accepted for publication after some revision. First, quantum-like approach is an extensively studied subject in recent years, for instance in ref. [1]. Classical neural network approach has been used in gravitational waves [2-3]. On the other hand, quantum networks have been proposed and studied [4-6]. As for the double-slit experiment, it is worth noting that the interpretation that the wave-function is the quantum entity itself, rather than just a mathematical description [8]. It can be tested in the encounter-delayed choice experiment in ref. [7], and upto now, only this interpretation explains the encounter-delayed choice experiment [8]. [1] Luo H M, Lin W, Chen Z C, et al. Extraction of gravitational wave signals with optimized convolutional neural network[J]. Frontiers of Physics, 2020, 15(1): 14601.
[2] Wang G Y, Long G L. Entanglement purification for memory nodes in a
quantum network[J]. SCIENCE CHINA Physics, Mechanics & Astronomy, 2020, 63(2): 220311.
Reviewer 2 Report
In this work, authors propose a novel formalism for defining probabilistic inference in quantum-like Bayesian networks. Their motivation is to define a consistent method for obtaining normalised probabilities from wave functions.
They have grounded their approach in the Law of Balance and the Law of Maximum Uncertainty (or Entropy). The presented derivations are very clear and easy to follow. Importantly, authors slowly introduce the reader to basic concepts Bayesian networks and their re-formulation in Quantum formalisms. This part of the paper is excellent.
However, I do have several concerns regarding empirical validation:
1. The actual experiment could be explained better. How many participants were involved in various studies? Did they perform a sequence of decisions or only one decision? How were the values in table one and two obtained? Is that mean or median probability over a group of participants. I find it surprising that if people know that other player defected they do not defect always. Why is that?
2. When comparing predictions of different models and observed data, summary statistics would be helpful. Currently, it is difficult to judge which model better fits the data. On a first look, it seems that DH is the closest to observed probabilities. What would be here prediction uncertainty? How can these models be used to explain between-subject variability and subject-specific priors and other model parameters?
3. I find it quite interesting that quantum effects disappear for very small prior probabilities (as shown in Fig 6 and 7). This seems to be a very clear prediction which could be experimentally tested. Are there any empirical data which could support this result?
4. Finally, my loudest critic would be that authors put too much confidence in the relevance of these results for understanding human behaviour without actually demonstrating empirically that the formalism they propose captures certain phenomena better than simpler models coming from classical Bayesian formalism and decision-making, e.g. by using model comparison and prediction of subject specific responses. Human behaviour deviating from the predictions of expected utility theory implies that the expected utility theory is potentially based on wrong assumptions and not that the people are irrational. The EUT has been strongly criticised in recent years, see for example --- Peters, Ole. "The ergodicity problem in economics." Nature Physics 15.12 (2019): 1216-1221. Similarly, behaviour deviating from ideal (Bayesian) observer implies that the generative model used to describe the behaviour is inappropriate and that potentially people have a different representation of the task than what experimented assumes. There are numerous examples in decision-making literature which show that with proper training, people start to behave closer to the predictions of an idealised model. Hence, I would like to suggest to the authors to shape their claims slightly differently and to at least acknowledge other (non-quantum like) explanations of why human behaviour appears irrational in various situations, see for example Schwartenbeck, Philipp, et al. "Optimal inference with suboptimal models: addiction and active Bayesian inference." Medical hypotheses 84.2 (2015): 109-117.
In what follows, I list various minor comments and errors that I noticed in the paper. Minor comments:
Abstract
1 . The authors mention both the low of maximum entropy and the low of maximum uncertainty. Sticking to one name would be less confusing.
2. The authors claim that their approach "shows what the decision-maker actually chooses, instead of normatively prescribing what the decision-maker should choose." Behavioural models are always made to predict what decision-maker chooses; but they can fit badly to the actual behaviour, implying that the assumptions used to derive the model are wrong. I am not sure what would be a model that prescribes choices. Could the authors clarify this a bit?
Probabilistic Inference in Bayesian Networks
1. line 104 - missing reference
2. eq 45 - an inversion of cosine got lost on the second line
Empirical validation
1. lines 300 and 305 - missing table number
2. line 313 - double "the"
3. Equation 67 is a bit confusing as the reader here expects x to be a categorical and y a binary variable, which is also the case for the next equation 68.
Conclusion
1. "Although we cannot test the proposed approach in more complex decision problems due to the nonexistence of data". Numerous experiments have investigated more complex problems, such as sequential decision making and planning. Both in the context of binary stimuli and responses and the categorical stimuli and response. What type of data is precisely missing in the literature in authors opinion?
Round 2
Reviewer 1 Report
I can recommend its acceptance.
Reviewer 2 Report
The authors have answered all my comments and concerns.